# Vancomycin-Loaded Furriness Amino Magnetic Nanospheres for Rapid Detection of Gram-Positive Water Bacterial Contamination

**DOI:** 10.3390/nano12030510

**Published:** 2022-02-01

**Authors:** Ahmed M. Azzam, Mohamed A. Shenashen, Mohamed S. Selim, Bayaumy Mostafa, Ahmed Tawfik, Sherif A. El-Safty

**Affiliations:** 1National Institute for Materials Science (NIMS), 1-2-1 Sengen, Tsukuba-shi 305-0047, Ibaraki-ken, Japan; ah.azzam@tbri.gov.eg (A.M.A.); moh.selim_chem2006@yahoo.com (M.S.S.); 2Environmental Research Department, Theodor Bilharz Research Institute (TBRI), Imbaba, Giza 12411, Egypt; bbmostafa@outlook.com; 3Egyptian Petroleum Research Institute (EPRI), Nasr City, Cairo 11727, Egypt; 4Water Pollution Research Department, National Research Centre (NRC), Dokki, Giza 12622, Egypt; prof.tawfik.nrc@gmail.com

**Keywords:** magnetic nanospheres, bacterial pollution, detection, vancomycin

## Abstract

Bacterial pathogens pose high threat to public health worldwide. Different types of nanomaterials have been synthesized for the rapid detection and elimination of pathogens from environmental samples. However, the selectivity of these materials remains challenging, because target bacterial pathogens commonly exist in complex samples at ultralow concentrations. In this study, we fabricated novel furry amino magnetic poly-L-ornithine (PLO)/amine-poly(ethylene glycol) (PEG)-COOH/vancomycin (VCM) (AM-PPV) nanospheres with high-loading VCM for vehicle tracking and the highly efficient capture of pathogens. The magnetic core was coated with organosilica and functionalized with cilia. The core consisted of PEG/PLO loaded with VCM conjugated to Gram-positive bacterial cell membranes, forming hydrogen bonds with terminal peptides. The characterization of AM-PPV nanospheres revealed an average particle size of 56 nm. The field-emission scanning electron microscopy (FE-SEM) micrographs showed well-controlled spherical AM-PPV nanospheres with an average size of 56 nm. The nanospheres were relatively rough and contained an additional 12.4 nm hydrodynamic layer of PLO/PEG/VCM, which provided additional stability in the suspension. The furry AM-PPV nanospheres exhibited a significant capture efficiency (>90%) and a high selectivity for detecting *Bacillus cereus* (employed as a model for Gram-positive bacteria) within 15 min, even in the presence of other biocompatible pathogens. Moreover, AM-PPV nanospheres rapidly and accurately detected *B. cereus* at levels less than 10 CFU/mL. The furry nano-design can potentially satisfy the increasing demand for the rapid and sensitive detection of pathogens in clinical and environmental samples.

## 1. Introduction

Pathogenic bacteria cause environmental contamination and serious diseases and have become a significant public health burden over time [1]. These pathogens responsible for many diseases, such as septicemia, meningitis, gastroenteritis, and abortion, which are all associated with a high mortality rate [2]. *Bacillus cereus* (*B. cereus*) is one of the most prevalent human pathogens in the environment, and it is a foodborne spore-forming bacterium causing food poisoning. *B. cereus* can cause many gastrointestinal disorders such as vomiting and diarrhea due to toxins production [3]. The development of a rapid, sensitive, and selective method for pathogen detection is vital to improve the treatment of diseases [4]. Many studies have investigated the rapid and highly selective detection of bacterial pathogens. However, highly accurate and satisfactory culture-based assays without pre-enrichment have not been established yet for detecting bacteria at ultralow concentrations in clinical or environmental specimens. Existing detection methods require a considerable amount of time [5]. Samples should be concentrated by minimizing the samples volumes for the rapid and dependent capture of pathogenic bacteria [6]. Immunological methods with antigen–antibody reaction have also been widely used in pathogen detection but require specific antibodies and complex technicians. Nucleic acid amplification-based methods show the potential for bacterial classification. PCR is a precise method which is able to detect one bacterium, but in case of pure DNA in complex environmental samples, it cannot guarantee satisfactory specificity. Methods for the rapid, specific, and sensitive detection of bacteria remain an interesting research topic, because the bacterial environment is excessively complex; moreover, the present purified techniques take a very long time and have a minimum reproductive rate [7,8]. Nanotechnology beats all these problems and implements a direct detection of target bacteria in complex environments in a short time. Many studies concerning nanotechnology-based biosensors showed promising outcomes for biological compounds and pathogenic agents based on silica, alumina, carbon nanotubes, magnesium hydroxide, silver, magnetic, and mesoporous nanomaterials [9,10,11,12,13,14,15].

The bio-application of magnetic iron oxide nanoparticles in the analysis field was recently demonstrated due to their attractive impacts, high loading capacity, and ease of effective modification [16,17,18,19]. For example, magnetic nanoparticles (MNPs) are used as a carrier for purification/separation [20]. Jo et al. used nano-MNPs for catching free cancer cells [21]. Kuo et al. [22] reported biofunctional MNPs for pathogen separation and concentration. MNPs are also utilized for purifying target bacteria and increasing their count to allow the easy bacterial identification [23].

Iron MNPs are functionalized with certain molecules, such as amine groups [24], antibodies [25], bacteriophages [26], nucleic acids [8], or peptides [22], for capturing bacterial cells. These functional groups can improve the sensitivity of pathogen detection based on the pre-enrichment of MNPs but exhibit limitations, such as high cost, low stability, and low selectivity, which impedes their application in the diagnosis of bacterial pathogens. Small, highly stable, and low-cost particles could be an attractive alternative to MNPs [27]. Broad-spectrum antibiotics are standard particles due to their high binding capacities to target bacterial cell membranes [28].

In this work, we reported novel fabricated furry multivalent amino magnetic core/organically functionalized multi-shells of poly-L-ornithine (PLO)/amine-poly(ethylene glycol) (PEG)-COOH/vancomycin (VCM) nanospheres (AM-PPV NSs) for the highly selective diagnosis and rapid separation of Gram-positive bacteria in complex environmental and clinical samples, as shown in Figure 1. PLO works as a linker between AM-NSs and VCM due to the presence of the amine group and improves the high loading capacity of the latter to provide multiple active sites. Thus, this orientation of AM-PPV NSs can satisfy the increasing demand for rapid pathogen diagnosis due to the following:
(I)The VCM corona on MNPs provides many active sites for the high selectivity of Gram-positive bacteria in mixed specimens;(II)The PLO and PEG polymers on AM-NSs provide elastic cilia, excellent scattering, and improved VCM enrichment efficiency; and(III)High concentrations of VCM on the AM shell qualify NSs to effectively attach to the target bacterial cell membrane.

The capacity and selectivity of NSs for *B. cereus* (employed as a model) were assayed to evaluate the feasibility of furry AM-PPV NSs. *B. cereus* is a rod-like Gram-positive bacterium that is prevalent in the environment. *B. cereus* is the main cause of food poisoning and other dangerous and fatal infections in the gastrointestinal tract [29]. The detection limit of AM-PPV NSs against *B. cereus* was determined. Moreover, the removal capacity and selectivity of the developed materials were tested. The fabricated furry magnetic NSs exhibited an efficient enrichment and a satisfactory purity and thus can be used to detect pathogens in environmental and clinical samples.

## 2. Materials and Methods

### 2.1. Materials and Reagents

All media and chemicals used in the present study were in analytical grade and used without further purification. 1-(3-dimethylaminopropyl)−3-ethylcarbodiimide hydrochloride (EDC), (3-Aminopropyl)triethoxysilane (APTES), N-hydroxysuccinimide (NHS), VCM hydrochloride, PLO (Mw: 15–30 kDa), amine-poly(ethylene glycol)-carboxymethyl (amine-PEG5000-COOH), 2-(N-morpholino) ethanesulfonic acid (MES), NaCl, KCl, Na_2_HPO_4_, KH_2_PO_4_, and fluorescence dye Hoechst 33258 were obtained with a 99% purity from Sigma-Aldrich (St. Louis, MI, USA). Nutrition media used in the present study were obtained from HiMedia Laboratories Ltd. (Mumbai, India) and the API 20E system (BioMerieux, Marcy-l’Étoile, France). *B. cereus*, *Pseudomonas aeruginosa* (*P. aeruginosa*), *Escherichia coli* (*E. coli*), and *Klebsiella pneumonia* (*K. pneumonia*) were obtained from Environmental Research Department, Theodor Bilharz Research Institute, Giza, Egypt. All aqueous solutions were prepared by Milli-Q water.

### 2.2. Furry AM-PPV NSs Fabrication Steps

#### 2.2.1. Synthesis of AM-NSs

Hydrophilic Fe_3_O_4_ NPs were fabricated by a simple alkaline deposition method [30]. The Fe_3_O_4_ NSs were synthesized by adding Fe^2+^ into an alkaline solution at 90 °C dropwise, in the presence of citrate and sodium nitrate. The obtained black precipitate was washed with water and ethanol several times and then dried at 50 °C. The hydrophilic MNSs were easily dispersed in ethanol and used as seeds in the next step. The formation of the amino shell was fabricated by dispersing 0.3 g of MNSs in 4 mL DW then and 600 mL of ethanol (99.99%) under sonication for 45 min. After that, 10% (*v/v*) APTES in an aqueous solution (80 mL) and glycerol (40 mL) were added to the above MNSs solution and stirred for 2 h in a nitrogen atmosphere at 90 °C. Then, the solution was cooled to room temperature. The NSs were washed with Milli-Q water and ethanol. The prepared AM-NSs were stored in Milli-Q water.

#### 2.2.2. Synthesis of VCM-PEG

VCM (30 mg) was mixed with NHS (60 mg) and EDC (80 mg) in 400 μL of dimethyl sulfoxide. The mixture was purified by recrystallization in diethyl ether. The purified VCM was stirred with amine polyethylene glycol carboxymethyl (H2N-PEG-COOH) (5.0 kDa, 60 mg) for 6 h in an alkaline solution (NaOH, pH 8.0). The VCM-PEG product was collected by ultrafiltration centrifugation (Figure 2A). The formation of the VCM-PEG molecules was confirmed using UV–VIS spectrometry with a UV–VIS spectrophotometer model, i.e., Solidspec-3700 model (Shimadzu, Kyoto, Japan).

#### 2.2.3. Synthesis of AM-PLO NSs

Equal volumes (400 μL) of PLO hydrobromide (50 µg/mL) and a NaOH solution (pH 8.0) were mixed at room temperature (RT), and ditert-butyl dicarbonate (30.6 mg) was added to form Boc-N-PLO precipitates, which were collected by centrifugation. AM-NSs (4 mg) were washed with a NaOH solution (0.01 M) and then mixed with Boc-N-PLO, NHS (4.4 mg), and EDC (2.2 mg) in DMSO (800 μL). The AM-PLO-N-Boc was washed several times with saturated NaCl through magnetic separation. After that, to remove the N-Boc-protecting group, trifluoroacetic acid (2.0 mL) and methylene chloride (2.0 mL) were added to functionalized NSs and kept on ice for 60 min (Figure 2B).

#### 2.2.4. Fabrication of Furry AM-PPV NSs

The fabrication of furry AM-PPV NSs was made by mixing EDC (6.5 mg) and NHS (13 mg) to AM-PLO and PEG-VCM solutions and shaking for 30 min. The AM-PPV NSs were magnetically separated, washed and stored in MES (30 mM, pH 6.0). The fabrication process of furry AM-PPV NSs is shown in Figure 2C.

### 2.3. Structural Characterization of Furry AM-PPV NSs

The structural features of the AM-PPV NSs were investigated via field-emission scanning electron microscopy (FE-SEM, JEOL model 6500, Tokyo, Japan). Before insertion into the chamber, the AM-PPV NSs powder was ground and fixed onto a specimen stub using a one double-sided carbon tape. Then, a 10 nm Pt film was coated via anion sputtering (Hitachi E-1030, Tokyo, Japan) at room temperature to obtain high-resolution micrographs. Before sputtering deposition, the Pt target (4 nm in diameter; purity: 99.95%) was sputter-cleaned in pure Ar. The sputtering deposition system used for the experiments consisted of a stainless steel chamber, evacuated down to 8 × 10^−5^ Pa with a turbo molecular pump backed up by a rotary pump. The Ar working pressure (2.8 × 10^−1^ Pa), the power supply (100 W), and the deposition rate were kept constant throughout these investigations. Moreover, to record the SEM micrographs of the AM-PPV NSs sample better, the scanning electron microscope was operated at 20 keV. ED-STEM was carried out at a camera length of 40 cm and a spot size of 1 nm. In the ED-STEM and FE-SEM characterization, the AM-PPV NSs were dispersed in an ethanol solution using an ultrasonic cleaner and then dropped on a copper grid.

X-ray photoelectron spectroscopy (XPS) was conducted using a PHI Quantera SXM (ULVAC-PHI) (Perkin-Elmer Co., Waltham, MA, USA) with monochromated AlKα radiation (1.5 × 0.1 mm, 15 kV, 50 W). Wide-angle powder X-ray diffraction (XRD) patterns were measured using an 18 kW diffractometer (Bruker D8 Advance, Billerica, MA, USA) with monochromated CuKα radiation. The sample measurement was repeated three times under rotation at various angles (15°, 30°, and 45°). The diffraction data were analyzed using the DIFRAC plus Evaluation Package (EVA) software with the PDF-2 Release 2009 databases provided by Bruker AXS. The standard diffraction data were identified according to the databases of the International Centre for Diffraction Data (ICDD).

### 2.4. Bacteria Sample Preparation

Bacterial pure strains were cultured in a shaker incubator in Luria broth (LB) media at 180 rpm and 37 °C for 12 h. During the exponential growth phase, the bacterial growth was centrifuged at a high speed, and then, the bacteria were collected and washed by PBS (0.1 M). To form a seed culture suspension, the concentration was adjusted to OD_600_ = 1 using a spectrophotometer (Spectronic 20D, Thermo Scientific, Waltham, MA, USA), which was equivalent to the bacterial concentration of 1 × 10^8^ CFU/mL.

### 2.5. Electronic Microscopy Imaging for the Captured B. cereus

The *B. cereus* seed culture was washed and resuspended using an autoclaved saline solution (0.9%), and 50 μL of AM-PPV NSs (5 mg/mL) were added to the bacterial suspension (1 mL, 1 × 10^3^ CFU/mL) on a shaker for 15 min. After magnetic separation, the conjugated *B. cereus* bacterial cells with AM-PPV NSs were examined using a transmission electron microscope (EM 208S Philips, Amsterdam, The Netherlands) at 80 kV and a scanning electronic microscope (HITACHI S-4800, Tokyo, Japan).

### 2.6. B. cereus Fluorescence Imaging

*B. cereus* bacterial cells at different concentrations (1 × 10^2^–1 × 10^8^ CFU/mL) were incubated with AM-PPV NSs and AM-VCM (5 mg/mL) in the presence of the diluted Hoechst 33258 dye for 15 min at RT under faint light [31]. The attached *B. cereus* bacterial cells and original solutions were examined and imaged with fluorescence microscopy at 352 nm.

### 2.7. The Plate Count Method for the Capture Capacity of AM-PPV NSs

AM-PPV NSs and AM-VCM NPs were added (5 mg/mL) to different *B. cereus* concentrations (1 × 10^2^–1 × 10^8^ CFU/mL) to test the capture capacity of AM-PPV NSs against AM-VCM for capturing Gram-positive bacterial cells. After incubation on the shaker for 15 min, the NSs were magnetically separated, and then supernatants solutions were cultivated on nutrient ager plates and incubated at overnight at 37 °C to detect the capacity of AM-PPV NSs for capturing bacterial cells. The corresponding capture efficiencies were calculated based on the following equation:
Capture efficiency (%) = [O − L]/O × 100,(1)
where O denotes the number of the original bacteria, and L indicates the remaining bacterial numbers after the separation from the NS-enriched *B. cereus*.

### 2.8. Specificity of AM-PPV NSs to Gram-Positive Pathogens

AM-PPV NSs (5.0 mg/mL) were added to the mixed water sample containing different bacterial species of G-positive (*B. cereus*) and G-negative ones (*P. aeruginosa*, *E. coli*, and *K. pneumonia*) at a concentration 1 × 10^4^ CFU/mL and incubated on a shaker for 15 min. After magnetic separation, AM-PPV NSs were washed three times with PBS, then spread on nutrient ager plates and incubated at overnight at 37 °C. The remaining water sample was cultivated on nutrient agar plates. After incubation, the growths of the conjugated bacterial species and the remaining ones were examined and identified using an Analytical Profile Index (API) 20E system.

## 3. Results and Discussion

### 3.1. Strategy Mechanism of the Furry NS–High-Loading VCM

Recently, more simple, sensitive, and cost-effective methods were progressed for the detection of pathogens [32,33,34]. In addition, highly sensitive and inexpensive techniques have been developed for bacterial capture [1,35]. Moreover, the outcomes are constantly unsatisfactory, primarily because pathogens commonly exist in complex samples in the environment [36]. To avoid this issue, VCM conjugated with magnetic nanoparticles was used for the detection of pathogens [37]. However, due to the poor loading of nanomaterials with VCM, the results remained impractical. In this study, we designed a “furry” ligand-functionalized AM-NS for the rapid and selective separation of Gram-positive bacterial species by favorable magnetic nanocomposites from samples of different types (Figure 1). Our novel design of furry NSs includes three steps, i.e., multi-conjugation, aggregation, and magnetic separation, to supply many characteristics as following:
(i)VCM is able to form a strong five-point hydrogen bonding with terminal groups of the cell wall structure of bacterium [17];(ii)Furry NSs have more rapid association kinetics with bacterial cells than direct AM-VCM NSs; thus, incubation times can be considerably shortened;(iii)The “furry” ligands can be easily fabricated and modified with MNSs, thereby improving the loading capacity of VCM on NSs and dispersibility in solutions and working as a vehicle-tracking building (VTB) for the target bacteria (Gram-positive).

The “furry action” effect of AM-PPV NSs could improve the loading capacity of VCM compared with that of direct AM-VCM NSs, which makes it more effective and accurate in the selective detection of most pathogens. The construction of furry AM-PPV NSs is schematically shown in Figure 2, and the furry AM-PPV NSs were fabricated based on the following:
(i)The activation of the carboxylate group of VCM by EDC/NHS, and amine-PEG-carboxymethyl was coordinated to VCM to form VCM-PEG;(ii)The immobilization of PLO through the carboxylate group on the amine-terminated MNSs to create AM-PLO through carbodiimide chemistry;(iii)The terminal −COOH (carboxylate group) of the VCM-PEG molecule was bonded with −NH_2_ (amino group) of AM-PLO, forming furry AM-PPV NSs.

### 3.2. Characterization of the Structural Features of Furry NSs

The FE-SEM images of AM-NSs are presented at low (Figure 1A) and high (Figure 1B) magnifications. The FE-SEM micrographs provided a clear evidence of the well-controlled morphology of AM NSs, with an average particle size of 56 nm. The image also showed the connection and overlapping of the NSs. The STEM images showed the surfaces of the AM and furry AM-PPV NSs (Figure 1C,D); it was relatively rough in AM-PPV NSs with an approximately size of 12.4 nm and contained an additional hydrodynamic layer of PLO/PEG/VCM, which provided additional stability in suspensions [38]. From Figure 1C, we can see that AM was heavily surrounded by the furriness from PLO/PEG/VCM. This improved the rapid separation with the help of a magnetic separator in only 15 min, through this furriness that was combined with the D-Ala-D-Ala on the surface of Gram-positive bacterial cell membranes by a five-point hydrogen bonding.

The WA-XRD (Wide Angle X-Ray Diffraction) patterns for the fabricated magnetic NSs were scanned within the 2*θ* range of 10°–80°. The characteristic peaks for Fe_3_O_4_ present in six diffraction angles of 2*θ* = 30.3°, 35.6°, 43.3°, 53.6°, 57.3°, and 62.9° were obtained in all tested materials (Figure 1E). These peaks were characterized as (220), (311), (400), (422), (511), and (440), respectively, which agrees totally with JCPDS card: 019-0629. The WA-XRD results revealed that magnetic NSs existed in all tested materials. Hence, AM, AM-PLO, and AM-PPV NSs possessed magnetism, so they could be directly removed from the water by a strong magnet easily [35].

Figure 2A confirms the successful conjugation between PEG and VCM, where the peak of VCM (20 µg/mL) was presented with a slight offset, which indicated the successful conjugation of VCM with PEG. PLO with an amine group in each repeat unit was formed to raise the capacity of concentration of VCM and upgrade the capture of the target bacterial cells efficacy. The absorption peak of AM-PPV furriness exhibited a high shift compared with that of VCM (Figure 2B), confirming a successful conjugation. These results confirmed the high loadings of VCM on the AM, forming AM-PPV NSs [39]. The successful synthesis of furry NSs also depended on the VCM loading quantity.

The VCM concentration was estimated to be 72.2 µg/mL of AM-PPV NSs based on the UV spectrometric method (Figure 3). This high loading of VCM on the AM-PLO NSs increased the selectivity capture efficiency of AM-PPV NSs for the target Gram-positive bacterial species, where high VCM loadings contributed to a multivalent interaction between the NSs and the terminal (D-alanyl-D-alanine) of the Gram-positive bacterial cell membrane through a five-point hydrogen bonding, as shown in Figure 1.

### 3.3. Functional Analysis of Furry AM-PPV NSs

The function of the NSs is a foundation of success that affects the performance of furriness-like enrichment AM-PPV NSs. The capture capacity of AM-PPV NSs for *B. cereus* was spotted through TEM and SEM analyses. As shown in Figure 4A, *B. cereus* was heavily surrounded by the furry AM-PPV NSs. The pathogenic bacterium was rapidly and easily separated using a magnetic separator within 15 min only. These findings demonstrated the capture ability of furry AM-PPV NSs for *B. cereus* as a model of Gram-positive bacteria, where all Gram-positive bacterial species had the same structure of the outer cell wall, which reacted with our NSs. Therefore, they will give the same results with our synthesized NSs and can easily separate all Gram-positive bacterial species from mixed clinical and environmental samples for rapid diagnosis.

### 3.4. Capture Efficiency of Furry Magnetic NSs

A comparative experiment of AM-PPV NSs and direct VCM conjugated to amino magnetic NSs (AM-VCM) was conducted to evaluate the capture performance of the furry NSs for *B. cereus*. Bacteria with different concentrations of ranging from 1 × 10^2^ to 1 × 10^8^ CFU/mL were stained by Hoechst 33258, a blue fluorescent dye that tended to interact with AT-rich DNA in cells and then were observed by fluorescent microscopy, to directly visualize the capture results. In Figure 4B, lane a shows the stained *B. cereus* in the original solution, whereas lane b presents the conjugates of AM-PPV NSs and *B. cereus*. The fluorescent images clearly indicated the binding of AM-PPV NSs to *B. cereus*. For all experiment groups, the fluorescence intensities increased with the increasing *B. cereus* concentration. Compared with the original solution (lane a), lane b displayed stronger fluorescence intensities, especially at high concentrations, indicating the high capture efficiency of the furry NSs for the target bacterium due to the presence of PLO furriness surrounding the AM-NSs, which provided great opportunities for a high load of VCM and helped in the strong capture of the target bacteria.

### 3.5. The Trapping Capability of the AM-PPV NSs toward Pathogens against AM-VCM

The trapping capability was investigated through the culturing of the remaining bacterial cells in water after the magnetic removals of both AM-PPV NSs and AM-VCM on agar plates. Figure 5A shows that the count of the left bacteria with AM-VCM NSs (Figure 5A(b1–b4)) was more than that of furry AM-PPV NSs (Figure 5A(c1–c4)).

The data in Figure 5B indicated that at the 10^2^ CFU/mL concentration, the capture capacity percentage of the furry AM-PPV NSs was significantly higher (95% ± 5%) than that of the AM-VCM (65% ± 5%); moreover, at an ultralow concentration (10 CFU/mL), the capture capacity percentage of the AM-PPV NSs was 90% ± 10% and the capture capacity percentage of AM-VCM NSs reached only 70% ± 5%. However, the capture efficiencies at the bacterial concentration of 10^3^ CFU/mL were 60% ± 5% and 90% ± 5%, while at 1 × 10^4^ CFU/mL the capture efficiencies recorded were 50% ± 5% and 85% ± 5% for AM-VCM and AM-PPV NSs, respectively (Table 1). Consistent with these findings, furry NSs worked as efficient nanocarriers for high-loading VCM concentrations, which supports the dissociation constant and exhibits an enhanced affinity for bacteria, resulting in a higher capture efficiency [40].

### 3.6. Selectivity of Furry AM-PPV NSs for Gram-Positive Bacteria in the Mixed Sample

The mixed water sample containing G-positive (*B. cereus*) and G-negative (*P. aeruginosa*, *K. pneumoniae*, and *E. coli*) bacteria was used for testing the selectivity of AM-PPV NSs, where furry NSs were applied for the mixed sample to separate *B. cereus* (Gram-positive) bacteria from the sample by a magnet, when the remaining bacterial species in the samples were cultivated on nutrient agar plates to demonstrate the specificity of AM-PPV NSs. The bacterial growths only detected on the plates were for *E. coli*, *K. pneumoniae*, and *P. aeruginosa* (Gram-negative), whereas no growth of *B. cereus* was observed on the plates as shown in Figure 3. These results confirmed that through the subsequent specificity of furry AM-PPV NSs for the detection of Gram-positive bacterial species, the furry AM-PPV NSs can be applied to complex clinical and environmental bacterial samples for the detection of bacterial contamination and rapid separation diagnosis [17].

## 4. Conclusions

We successfully constructed novel furry magnetic high-loading VCM NSs with a high selectivity for rapid pathogen detection. Our designed AM-PPV NSs increased the loading of VCM, upgraded the capture capacity of bacterial cells and worked as a VTB for the target Gram-positive bacterial species, where PLO with a repeat amino group formed the corona of elastic cilia terminated with VCM molecules. VCM was able to form a strong five-point hydrogen bonding with terminal groups of the cell wall structure of the bacterium. Thus, the modified AM-PPV NSs obtained more dispersibility and biocompatibility and can be used in complex environmental and clinical samples. The use of AM-PPV NSs were more rapid than other previous methods, where ˃90% of *B. cereus* bacterial cells could be efficiently captured within only 15 min with a very low detection limit reaching an ultralow concentration of 10 CFU/mL. These properties and performance make AM-PPV NSs a potentially useful approach to the rapid diagnosis of Gram-positive bacterial pathogens in clinical and environmental samples.

## Data Availability

Data Availability Statement: Data can be available upon request from the authors.

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
