# Peer review of "Vancomycin-Loaded Furriness Amino Magnetic Nanospheres for Rapid Detection of Gram-Positive Water Bacterial Contamination"

_nanomaterials, 2022, doi:10.3390/nano12030510_

Round 1

Reviewer 1 Report

In this manuscript, Ahmed M. Azzam et al. developed furriness magnetic high loading vancomycin nanospheres for vehicle tracking and highly efficient capture of pathogens. VCM can be capable to form strong hydrogen bonding with terminal groups of the cell wall structure of bacterium. Results showed that the fabricated nanostructures can be used in complex environmental and clinical samples. I found that the work seems to be carefully carried out and the results are well explained. I recommend the manuscript be accepted by the Nanomaterials after minor revision. The detailed comments are given below.

1) What concentrations of materials were used for measurements in Figure 2?

2) Please include error bars in Figure 3.

3) Scale bar is missing in Figure 4B.

4) Please give a comment on the advantages of this method as compared to the previously reported ones.

Author Response

Author response upon the Reviewers’ Comments

 First of all, we thank the reviewers for their efforts and valuable comments. We appreciate their concern and recommendation upon our work. Per of this respectful vision, we thoroughly amended our manuscript taking all points raised in our consideration.

Reviewer #1 

Comments and Suggestions for Authors:

In this manuscript, Ahmed M. Azzam et al. developed furriness magnetic high loading vancomycin nanospheres for vehicle tracking and highly efficient capture of pathogens. VCM can be capable to form strong hydrogen bonding with terminal groups of the cell wall structure of bacterium. Results showed that the fabricated nanostructures can be used in complex environmental and clinical samples. I found that the work seems to be carefully carried out and the results are well explained. I recommend the manuscript be accepted by the Nanomaterials after minor revision. The detailed comments are given below.

We thank the reviewer for his valuable opinion about our manuscript and appreciate his review and comments that contributed to the improvement of the work under review. Furthermore, we would like to emphasize that the comments raised have been carefully addressed in the revised form.

Q1. What concentrations of materials were used for measurements in Figure 2?

A1. We thank the reviewer for his comment. Per of this comment, the concentrations of material used in this study was 20 µgmL-1 (Please see the confirmation on the highlighted manuscript).

Q2.  Please include error bars in Figure 3.

A2. We appreciated the reviewer’s comment. Per of this comment, the figure has been revised and redesign, please see the modified figure as follows:

Q3. Scale bar is missing in Figure 4B.

A3. We appreciated the reviewer’s comment. Per of this comment, the Scale bars has been added (50 μm), please see the modified figure as follows:

Figure 4. (A) SEM micrographs of control (a) and captured Bacillus cereus (b), which conjugated with AM-PPV nanospheres; TEM micrographs of B. cereus control (c) and captured (d) showed the attached AM-PPV NSs on the bacterial cell membrane, and (B) Fluorescence detection of captured B. cereus bacteria by furriness amino magnetic core/organically functionalized multi-shells nanosphere (AM-PPV) and direct AM-Vancomycin (AM-V). The concentrations of B. cereus stained by fluorescence dye ranges from 1×102 to 1×108 cfu mL-1. Scale bars 50 μm.

Q4. Please give a comment on the advantages of this method as compared to the previously reported ones.

A4) We thank the reviewer for his comment. The problem of mixed samples of both Gram bacterial species, which it takes more time for separation between them using conventional methods (more than 48 hours). As mentioned in manuscript (Highlight with yellow and Table 1).

  1. The advantage of the designed material related to its use as a vehicle tracking building (VTB) for target bacterial cells (Gram-positive bacteria) due to high loading of vancomycin on terminals furriness.
  2. Moreover, our method has good performances toward removing of all Gram-positive bacterial species from mixed samples within only 15 min.
  3. Table 1 Showed comparison between efficiency of AM-VCM (Old technique) and AM-PPV nanospheres (Present our material) for capture Bacillus cereus bacteria, revealed that our material has highest capacity for capture bacteria.

Finally, we would like to acknowledge the efforts of the reviewers and the valuable comments that improved our manuscript thoroughly. Hope this amendment will take your consideration.

Reviewer 2 Report

In this manuscript, the authors prepared vancomycin-loaded furriness amino magnetic nanospheres for rapid detection of Gram-positive water bacterial contamination. The topic is meaningful, however, the current content of the manuscript does not fit the publishable standards of nanomaterials journal, so I would suggest a major revision is needed for this paper. Please see my comments below for details.

  1. There are many abbreviations in the article, representing a range of materials or intermediate materials, which can easily be confusing. Please redraw Figure 1 to put all the material acronyms involved together, so that the reader can quickly understand the framework and ideas of the whole paper.
  2. The method of material preparation should be put together and followed by the information on material characterization. Specifically, section 2.5 should be moved to the front and combined with 2.24. And 2.3 should have the word “characterization” instead of “feature”.
  3. The contents of Figure 2 and Figure 3 should be integrated in a suitable way.
  4. The resolution and quality of all figures need to be further improved. In addition, the structural information of the involved molecules in Figures 1 to 3 is not clear enough and it would be better to draw the chemical structure formula to illustrate it, as the ball-and-stick model is not very suitable and clear.
  5. The title of the article deals with Gram-positive bacteria, but only one type of bacteria is used in the article, whether the material is also effective against other Gram-positive bacteria.
  6. Can the material be used multiple times.

Author Response

Author response upon the Reviewers’ Comments

First of all, we thank the reviewers for their efforts and valuable comments. We appreciate their concern and recommendation upon our work. Per of this respectful vision, we thoroughly amended our manuscript taking all points raised in our consideration.

Reviewer #2 

Comments and Suggestions for Authors:

In this manuscript, the authors prepared vancomycin-loaded furriness amino magnetic nanospheres for rapid detection of Gram-positive water bacterial contamination. The topic is meaningful, however, the current content of the manuscript does not fit the publishable standards of nanomaterials journal, so I would suggest a major revision is needed for this paper. Please see my comments below for details.

 We thank the reviewer for this good opinion about our manuscript. We would like to emphasize that all comments have been considered and notified in our manuscript.

  • There are many abbreviations in the article, representing a range of materials or intermediate materials, which can easily be confusing. Please redraw Figure 1 to put all the material acronyms involved together, so that the reader can quickly understand the framework and ideas of the whole paper.

We thank the reviewer for his comment, although all abbreviations were mentioned in the manuscript, but all abbreviations in figure 1 were mentioned in the figure caption as follows:

Figure 1. FE-SEM images with low- (A) and high (B) magnifications of amino magnetic (AM) nanospheres, STEM image with high-magnification of (C) AM NSs, and (D) AM coated with poly-L-ornithine/PEG/Vancomycin (AM-PPV) nanospheres. (E) WA-XRD of magnetic nanosphere (Fe3O4), amino magnetic NSs (AM), AM-poly-L-ornithine (AM-PLO), and AM-PPV.

  • The method of material preparation should be put together and followed by the information on material characterization. Specifically, section 2.5 should be moved to the front and combined with 2.24. And 2.3 should have the word “characterization” instead of “feature”.

 We thank the reviewer for this right comment; this section was moved, and the word “characterization” was added instead of “feature” in section 2.3

  • The contents of Figure 2 and Figure 3 should be integrated in a suitable way.

 We thank the reviewer for this comment; figure 3 has been modified, please see the modified version as follows:

Figure 3. Determination of vancomycin (VCM) loading on AM-PPV nanospheres. (A, B) Calibration curve of VCM at 280 nm. The concentration ranges from 9.25 to 185 μɡ mL-1, (C) Absorption of AM-PPV NSs in the UV-vis spectrum, and (D) Statistical results of C at 280 nm.

  • The resolution and quality of all figures need to be further improved. In addition, the structural information of the involved molecules in Figures 1 to 3 is not clear enough and it would be better to draw the chemical structure formula to illustrate it, as the ball-and-stick model is not very suitable and clear.

 We thank the reviewer for this right comment; the Schemes 2 and 3 were integrated and all chemical structure formula was illustrated in new scheme.  

  • The title of the article deals with Gram-positive bacteria, but only one type of bacteria is used in the article, whether the material is also effective against other Gram-positive bacteria.

We appreciated the reviewer comment. As known, all Gram-positive bacterial species have the same structure, so during this study we an example of Gram-positive bacteria to confirm the material performances. In this regards, the abstract has been modified a follows:

 “These findings demonstrate the capture ability of furriness AM-PPV NSs for B. cereus as model of Gram-positive bacteria, where all Gram-positive bacterial species have the same structure of outer cell wall, which react with our nanosphere. So, they will give the same results with our synthesized nanosphere and can be easily separating all Gram-positive bacterial species from mixed clinical and environmental samples for rapid diagnosis.”

  • Can the material be used multiple times.

A3. We appreciated the reviewer’s comment. In fact, the reusability is recommended in the removal process. In this work to perform the recovery process, acidic solution should be used, which affected the functionalization of the designed the building structure of terminal furriness that is responsible for capturing bacteria, while the amino magnetic core of nanospheres will be more stable. In this regards, for multiple use, the amino magnetic core of nanospheres should be refunctionalized according to Scheme 2.

 Finally, we would like to acknowledge the efforts of the reviewers and the valuable comments that improved our manuscript thoroughly. Hope this amendment will take your consideration.

Reviewer 3 Report

The author of this article provide a quick and easy method to detect Galanz-positive bacteria. As claimed, this method is suitable for separating more than 90% of Galanz-positive bacteria, and is a potential medical wastewater detection technique. However, some issues still need to be improved before the article is considered for acceptance.

(1) According to Figure 1a, the size of the nanosphere is ~25 nm, but the size of the nanosphere  in Figure 1b is obviously larger than this size (even larger than 50 nm).  I suggest that the author should check if the scale bar is marked correctly.

(2) In my opinion, the evidence for the successful synthesis of AM-PPV nanospheres is far from enough. First and foremost, the author should at least provide the FTIR spectrum of PLO/PEG/VCM shell to confirm that the conjugation is successful. I also encourage the author to provide further NMR and MS data to make this result more convincing. Secondly, I suggest the author to provide EDS-Mapping and HRTEM image of AM-PPV nanospheres to further confirm the composition and crystal structure of the nanospheres.

(3) Many pictures (scheme 1, scheme 2, scheme 3, figure 1, figure 4, figure 5) have different degrees of stretching and deformation, which makes the article look unprofessional.

(4) There are many errors in the figures, such as the spelling error in Scheme 3 ("AM-PPV-Nanospher") and Figure 4B lacks a scale bar.

(5) In Figure 4, the use of icons such as "(A)" and "(a)" at the same time makes the entire screen confusing and difficult to understand. I suggest the author using (i), (ii).... instead of (a), (b).

Author Response

Author response upon the Reviewers’ Comments

First of all, we thank the reviewers for their efforts and valuable comments. We appreciate their concern and recommendation upon our work. Per of this respectful vision, we thoroughly amended our manuscript taking all points raised in our consideration.

Reviewer #3

The author of this article provide a quick and easy method to detect Galanz-positive bacteria. As claimed, this method is suitable for separating more than 90% of Galanz-positive bacteria, and is a potential medical wastewater `detection technique. However, some issues still need to be improved before the article is considered for acceptance.

 We thank the reviewer for this good opinion about our manuscript. We would like to emphasize that all comments have been considered and notified in our manuscript.

  • According to Figure 1a, the size of the nanosphere is ~25 nm, but the size of the nanosphere in Figure 1b is obviously larger than this size (even larger than 50 nm).  I suggest that the author should check if the scale bar is marked correctly.

 We thank the reviewer for his valuable observation. Per of the reviewer comments, the figure design and scales have been checked and modified. The well-controlled morphology of AM-PPV shows average particle size of 56 nm, in this regard, Please see the modified Figure 1 as follows:

Figure 1. FE-SEM images with low- (A) and high (B) magnifications of amino magnetic (AM) nanospheres, STEM image with high-magnification of (C) AM NSs, and (D) AM coated with poly-L-ornithine/PEG/Vancomycin (AM-PPV) nanospheres. (E) WA-XRD of magnetic nanosphere (Fe3O4), amino magnetic NSs (AM), AM-poly-L-ornithine (AM-PLO), and AM-PPV.

  • In my opinion, the evidence for the successful synthesis of AM-PPV nanospheres is far from enough. First and foremost, the author should at least provide the FTIR spectrum of PLO/PEG/VCM shell to confirm that the conjugation is successful. I also encourage the author to provide further NMR and MS data to make this result more convincing. Secondly, I suggest the author to provide EDS-Mapping and HRTEM image of AM-PPV nanospheres to further confirm the composition and crystal structure of the nanospheres.

 We appreciate your nice suggestion. Also, we acknowledge the efforts of the Reviewer and his valuable Comments and Suggestions that improved our manuscript. According to the reviewer's comment, we would like to confirm the outline regarding material confirmation in our study as follows:

 In section 2.2.4. :

“The fabrication of furriness magnetic core organically functionalized multi-shells (AM-PPV) nanospheres was made by mixing EDC (6.5 mg) and NHS (13 mg) to AM-PLO and PEG-VCM solution and shaking for 30 min. The AM-PPV nanosheres were magnetically separated, washed and stored in MES (30 mM, pH 6.0). “

First, the magnetic separated particles after washing are only particles AM-PLO that were conjugated with PEG-VCM and the unconjugated molecules of PEG-VCM were removed through washing, which indicating the successful synthesis of AM-PPV nanospheres.

Second, for confirmation loading of vancomycin (PEG-VCM) on the magnetic nanospheres (AM-PLO) UV-vis spectrometry was used in steeps:

  1. UV-vis spectrometry (Figure 2A) confirmed the successful conjugation between PEG and VCM, where the peak of VCM was presented with a slight offset, which indicates the successful conjugation of VCM with PEG.
  2. UV-vis spectrometry (Figure 2B) confirmed the successful conjugation between AM-PLO and PEG-VCM, wheret the absorption peak of AM-PPV NSs exhibits a high shift compared with AM and AM-PLO (Figure 2B) confirming successful conjugation.
  • UV-vis spectrometry (Figure 3) confirmed the high loadings of VCM on the AM, forming AM-PPV NSs, where the VCM concentration was determined using calibration curve of VCM at 280 nm (The concentration ranges from 9.25 to 185 μɡ mL-1), and the VCM conc. estimated to be 72.2 µg ml-1 of AM-PPV NSs based on the UV spectrometric method.
  1. Finally, the SEM and TEM images of captured bacteria showed that high concentration of AM-PPV NSs surrounding the bacterial cell, which improved the formation of H-bonds between Vancomycin terminal heads of AM-PPV with terminal (D-alanyl-D-alanine) of bacterial cell membrane through five-point hydrogen bonding.
  • Many pictures (scheme 1, scheme 2, scheme 3, figure 1, figure 4, figure 5) have different degrees of stretching and deformation, which makes the article look unprofessional.

 We thank the reviewer for this right comment; Schemes 2 and 3 were integrated into one new scheme, also the degrees of stretching and deformation were readjusted.

  • There are many errors in the figures, such as the spelling error in Scheme 3 ("AM-PPV-Nanospher") and Figure 4B lacks a scale bar.

We thank the reviewer for this right comment; Schemes 2 and 3 were integrated into one new scheme and all spelling errors were corrected, also the scale bar was added to Figure 4B (Scale bars 50 μm).

  • In Figure 4, the use of icons such as "(A)" and "(a)" at the same time makes the entire screen confusing and difficult to understand. I suggest the author using (i), (ii).... instead of (a), (b).

We thank the reviewer for this right suggestion. So, in Figure 4 we added (i), (ii) instead of (a), (b).

Finally, we would like to acknowledge the efforts of the reviewers and the valuable comments that improved our manuscript thoroughly. Hope this amendment will take your consideration.

Round 2

Reviewer 2 Report

The revisions are helpful, and the manuscript is obviously improved and can be accepted after appropriate language polish.

Reviewer 3 Report

It can be accepted in the current form.